# New Physicochemical Methodology for the Determination of the Surface Thermodynamic Properties of Solid Particles

**Tayssir Hamieh** [1,2]

1   Faculty of Science and Engineering, Maastricht University, P.O. Box 616,
    6200 MD Maastricht, The Netherlands; t.hamieh@maastrichtuniversity.nl; Tel.: +31-6-5723-9324
2   Laboratory of Materials, Catalysis, Environment and Analytical Methods Laboratory (MCEMA), Faculty of
    Sciences, Lebanese University, Hadath P.O. Box 6573, Lebanon

**Abstract:** The study of the surface thermodynamic properties of solid materials is primordial for the determination of the dispersive surface energy, polar enthalpy of adsorption and Lewis's acid base properties of solid particles. The inverse gas chromatography technique (IGC) at infinite dilution is the best surface technique for the determination of the surface physicochemical properties of materials. (1) Background: This paper was devoted to studying the surface properties of solid materials, such as alumina, titania and silica particles, using the IGC technique. (2) Methods: Different methods and molecular models, such as the spherical, cylindrical, Van der Waals, Redlich–Kwong, Kiselev and geometric models, were used to determine the London dispersive surface energy of solid surfaces. The Hamieh model was also used and highlighted the thermal effect on the surface area of solvents. (3) Results: The variations of the dispersive surface energy and the free energy of adsorption were determined for solid particles as a function of the temperature, as well as their Lewis's acid base constants. Alumina surfaces were proved to exhibit a strong Lewis amphoteric character three times more basic than acidic, titanium dioxide more strongly basic than acidic and silica surface exhibited the stronger acidity. (4) Conclusions: The new methodology, based on the Hamieh model, gave the more accurate results of the physicochemical properties of the particle surfaces.

**Keywords:** dispersive surface energy; free energy of adsorption; enthalpy and entropy of adsorption; Lewis's acid base parameters; Hamieh thermal model; work of adhesion

## 1. Introduction

In most industrial processes, such as adhesion, adsorption, polymer synthesis, dispersion, food manufacturing, pharmaceutical drugs, biomedicine, clays, composites, materials and nanomaterials, it is necessary to determine the physicochemical, surfaces and interfaces properties of solid materials. One of the most famous techniques that gives information on the surface properties of materials and nanomaterials is inverse gas chromatography (IGC). This technique is largely successful in determining the surface physicochemical properties of materials, such as the dispersive surface energy, the specific free energy of adsorption and the Lewis-acid base parameters. Using the IGC technique allowed observation of the interactions between oxides, polymers or polymers adsorbed on oxides and organic solvent systems [1]. It was proved that this technique is very precise, sensitive and more competitive in studying the heterogeneous surfaces of solid surfaces, their physicochemical properties [2] as well as to determine the dispersive surface energy of powdered materials [3,4]. This attractive technique has been used since 1970 to quantify the specific interactions and the surface properties and glass transition of polymers, copolymers, their blends and polymer films [5–18]. It was also used to determine the surface energy, the physicochemical properties and the Lewis's acid base of metals and metal oxides, minerals, clay minerals [19–31], silicas and porous materials [32–39], nanomaterials [40–46], pharmaceuticals and food products [47–54] and other materials [55–69].

Many methods have been proposed in the literature and used during the last fifty years. At the beginning, Sawyer and Brookman [70] found, in 1968, an excellent linearity of the logarithm of the net retention volume $Vn$ of an adsorbed solvent on a solid, as a function of the boiling point $T_{B.P.}$ of n-alkanes $lnVn = f(T_{B.P.})$. The separation method of the dispersive (or London) and polar (or specific) interactions between a solid substrate and a polar molecule was proposed by Saint-Flour and Papirer [12,13]. These authors used the representation of $RTlnVn$ versus the logarithm of the vapor pressure $P_0$ of probes:

$$RTlnVn = \alpha P_0 + \beta \tag{1}$$

where $R$ is the ideal gas constant, $T$ is the absolute temperature and the $\alpha$ and $\beta$ constants depend on the interface solid–solvent. The distance relating the representative point of $RTlnVn$ of a polar molecule to its hypothetic point located on the n-alkane straight line determined the specific free energy of adsorption $\Delta G_a^{sp}$. The variation of $\Delta G_a^{sp}$ versus the temperature led to the specific enthalpy $\Delta H_a^{sp}$ and entropy $\Delta S_a^{sp}$ of polar molecule adsorbed and, therefore, to the Lewis acid-base parameters. Five other IGC methods were proposed, to characterize the solid surfaces, a similar linearity to separate the two dispersive and polar components of the specific interactions. Two similar methods were used to determine the dispersive component $\gamma_s^d$ of the surface energy of the solid. These methods are given below:

1.   Dorris and Gray [71] first determined the $\gamma_s^d$ of solid materials by using the Fowkes relation [72] and correlating the work of adhesion $W_a$ to the free energy of adsorption $\Delta G_a^0$ using the following relation:

$$\Delta G_a^0 = \mathcal{N}a\, W_a = 2\mathcal{N}a\, \sqrt{\gamma_l^d \gamma_s^d} \tag{2}$$

where $a$ is the surface area of adsorbed molecule, $\gamma_l^d$ the dispersive component of the liquid solvent $\gamma_l^d$ and $\mathcal{N}$ is Avogadro's number.

Dorris and Gray introduced the increment $\Delta G^0{}_{-CH2-}$ of two consecutive n-alkanes $C_n H_{2(n+1)}$ and $C_n H_{2(n+1)}$:

$$\Delta G^0{}_{-CH2-} = \Delta G^0\left(C_{n+1}H_{2(n+2)}\right) - \Delta G^0\left(C_n H_{2(n+1)}\right) \tag{3}$$

By supposing the surface area of the methylene group, $a_{-CH2-} = 6$, independent from the temperature and the surface energy, $\gamma_{-CH2-}$ (in mJ/m$^2$) of –CH$_2$- is equal to:

$$\gamma_{-CH2-} = 52.603 - 0.058\, T\ (T\ in\ K)$$

Dorris and Gray [71] then deduced the value of $\gamma_s^d$ using Equation (3):

$$\gamma_s^d = \frac{\left[RTln\left[\frac{V_n\left(C_{n+1}H_{2(n+2)}\right)}{V_n\left(C_n H_{2(n+1)}\right)}\right]\right]^2}{4\mathcal{N}^2\, a^2{}_{-CH2-}\, \gamma_{-CH2-}} \tag{4}$$

2.   The method proposed by Schultz et al. [73], using the Fowkes relation [72], similarly gave the free energy of adsorption $\Delta G_a^0$ as a function of the geometric mean of the respective dispersive components of the surface energy of the liquid solvent $\gamma_l^d$ and the solid $\gamma_s^d$:

$$\Delta G_a^0 = RTlnVn + C = 2\mathcal{N}a\left(\gamma_l^d \gamma_s^d\right)^{1/2} + D \tag{5}$$

where $a$ is the surface area of the probes' supposed constant for all temperatures, and $C$ and $D$ are two constants depending on the used materials and the temperature. The variations

of the $RTlnVn$ versus the $2\mathcal{N}a\left(\gamma_l^d\right)^{1/2}$ of n-alkanes and polar molecules gave both the $\gamma_s^d$ and $\Delta G_a^{sp}(T)$ of the solid.

In previous studies, it was determined the dispersive component of many solid materials by using the various molecular areas of Kiselev, Van der Waals (VDW), Redlich–Kwong (R-K), Kiselev, geometric, cylindrical or spherical models [74–79].

3. The method deduced from the works of Sawyer and Brookman [70] used:

$$RTlnVn = AT_{B.P.} + B \tag{6}$$

where $A$ and $B$ are two constants. This method gave the specific free energy and the acid base properties.

4. The method of the deformation polarizability $\alpha_0$ proposed by Donnet et al. [80]. They proposed the following relation:

$$RTlnVn = E(h\nu_L)^{1/2}\,\alpha_{0,\,L} + F \tag{7}$$

where $\nu_L$ is the electronic frequency of the probe, $h$ the Planck's constant and $E$ and $F$ are the constants of interaction.

5. Chehimi et al. [59] used the standard enthalpy of vaporization $\Delta H_{vap.}^0$ (supposed constant) of n-alkanes and polar molecules:

$$RTlnVn = I\Delta H_{vap.}^0 + J \tag{8}$$

where $I$ and $J$ are two constants. This method is similar to the Saint-Flour and Papirer method using $lnP_0$ and that of Sawyer and Brookman using $T_{B.P.}$.

6. The method of Brendlé and Papirer [2] used the concept of the topological index $\chi_T$; that is, a parameter considering the topology and the local electronic density in the polar probe structure. They gave the following relation:

$$RTlnVn = Mf(\chi_T) + N \tag{9}$$

where $M$ and $N$ are two adsorption constants.

In all previous cases, the determination of the $\Delta G_a^{sp}(T)$ of polar solvents versus the temperature will allow the deduction of the specific enthalpy $\left(-\Delta H_a^{sp}\right)$ and entropy $\left(\Delta S_a^{sp}\right)$ of the polar probes adsorbed on the solid surfaces by using Equation (1):

$$\Delta G_a^{sp}(T) = \Delta H_a^{sp} - \Delta S_a^{sp} \tag{10}$$

Knowing of $\Delta H_a^{sp}$ polar solvents, the two respective acid base constants, $K_A$ and $K_D$, of solids can be determined by Papirer following the relation [12,13]:

$$-\Delta H^{Sp} = K_A \times DN + K_D \times AN \tag{11}$$

That can be also written as:

$$\frac{-\Delta H^{Sp}}{AN} = K_A\frac{DN}{AN} + K_D \tag{12}$$

where $AN$ and $DN$, respectively, represent the electron donor and acceptor numbers of the polar molecule given by Gutmann [62] and corrected by Fowkes, $K_A$ and $K_D$ are the respective acid and base constant.

Since 1982, scientists have been interested in the determining the physicochemical properties of materials by using inverse gas chromatography at infinite dilution. Saint-Flour

and Papirer [12,13] first tried to separate the two dispersive and polar contributions of the Gibbs free energy of adsorption of polar solvents on solid substrates by using the notion of vapor pressure of organic molecules. Schultz et al. [73], Donnet et al. [80], Brendlé and Papirer [2] and other scientists [10,14,20,23,33,36,59] were also interested in determining the specific and dispersive properties of materials. There was no universal method or model to be used in IGC at an infinite dilution for an accurate characterization of solid particles. Even the Dorris–Gray relation [71] cannot give a precise value of the dispersive component of the solid surface of solid materials. Hamieh et al. [35,77,78,81] have nevertheless succeeded in carrying out a very precise determination of the second-order temperatures (such as the glass transition) of polymers, such as poly (methyl methacrylate) or pol (α-n-alkyl) methacrylate in the bulk phase or in the adsorbed state. The serious difficulty faced by researchers over 40 years was the problem of the diversity of the methods and models used to determine the surface properties of a solid material, such as the dispersive free surface energy, the specific free energy and the Lewis's acid-base constants without obtaining the same results. Indeed, we proved in several previous studies that the various models and methods of the IGC technique did not give the same results. On the contrary, the methods used gave results completely different from each other. The error percentage sometimes exceeds 100% from one method to another. Our previous works highlighted the important effect of the temperature on the surface area of organic solvents. We failed the methods proposed by Dorris–Gray [71] and Schultz et al. [73] that supposed a constant value of the surface area of organic molecules, and we proposed various relations of the surface area of probes depending on the temperature and corrected the values of the dispersive surface energy and polar properties of materials. Our new methodology was applied to the alumina particles and determined, with an excellent accuracy, the surface physicochemical properties of alumina. The thermal model [69] must be applied in the future for a good characterization of solid surfaces.

In this paper, we developed a new methodology for the determination of the physico-chemical, dispersive and polar properties of alumina particles by using the IGC technique at infinite dilution. We used all classical IGC methods and proposed to apply the new Hamieh's thermal model [69], which was proved to be more accurate than the other models in determining the specific free enthalpy and enthalpy of adsorption and the acid-base constants of the different materials. Eight molecular models of organic molecules were also used to calculate the dispersive component of the surface energy of solid particles.

## 2. New Methodology

### 2.1. Molecular Models

Many scientists [19–31] continued using the Fowkes relation to determine the London dispersive surface energy and the specific properties of the solid surfaces, as well as their Lewis acid-base parameters. They used the expression relating the free surface energy and retention volume to $\gamma_s^d$: $\Delta G_a^0 = RTlnVn + C = 2\mathcal{N}a \left( \gamma_l^d \gamma_s^d \right)^{1/2} + D$ by using the the surface area, *a*, of organic probes.

The surface areas of solvents used in the literature are those proposed by Kiselev and are the supposed constant for all temperatures. In order to prove that there are different ways to determine the surface areas of molecules depending on their geometry and their position during their adsorption on the solid surfaces, Hamieh et al. [79] proposed six new molecular models that allowed the calculation of the surface areas of organic molecules: the spherical (Sph.), geometric (Geom.), cylindric (Cyl.), Redlich–Kwong (R-K) and Van der Waals (VDW) models compared to the Kiselev results given in Table 1.

**Table 1.** Surface areas of n-alkanes (in Å$^2$) using the various molecular models: spherical (Sph.), geometric (Geom.), Redlich-Kwong (R-K), cylindric (Cyl.), Kiselev and Van der Waals (VDW).

| Cn | Sph. | Geom. | R-K | Cyl. | Kiselev | VDW |
|----|------|-------|-----|------|---------|-----|
| C5 | 36.4 | 32.9 | 36.8 | 39.3 | 45 | 47 |
| C6 | 39.6 | 40.7 | 41.3 | 45.5 | 51.5 | 52.7 |
| C7 | 42.7 | 48.5 | 46.4 | 51.8 | 57 | 59.2 |
| C8 | 45.7 | 56.2 | 50.8 | 58.1 | 63 | 64.9 |
| C9 | 48.7 | 64 | 54.5 | 64.4 | 69 | 69.6 |
| C10 | 51.7 | 71.8 | 58.2 | 70.7 | 75 | 74.4 |

All these molecular models were used for the determination of the London dispersive surface energy and the Lewis's acid base parameters of solid particles. The obtained results strongly depend on the chosen molecular model, and the difference between $\gamma_s^d$ of these models can sometimes reach three times the value obtained by the thermal model.

The other difficulty was the large dependency of the surface area and the surface tension of the solvents on the temperature. This will give wrong values of the thermodynamic properties of the materials. The correction of such superficial properties of materials must be performed in order to have the correct physicochemical behavior between solid surfaces.

*2.2. Hamieh's Thermal Model*

In a recent study, Hamieh [69] proved the dependency of the surface areas of molecules on the temperature. He gave the following relation of the surface area $a(n, T)$ of n-alkanes as a function of the temperature:

$$a(n, T) = \frac{69.939 \, n + 313.228}{(563.02 - T)^{1/2}} \tag{13}$$

By showing the failure of the Dorris–Gray method, which was largely used to determine the $\gamma_s^d$ of solids, this method considered the surface area $a_{-CH2-}$ of the methylene group equal to 6 Å$^2$ and constant for any used temperature. Hamieh [69] proved the non-validity of the Dorris–Gray method and gave the following expression of $a_{-CH2-}$ (in Å$^2$) as a function of the temperature $T$ (in K):

$$a_{-CH2-} = \frac{69.939}{(563.02 - T)^{1/2}} \tag{14}$$

Hamieh also gave the surface areas of polar molecules against the temperature by defining three new surface parameters: a first maximum temperature $T_{Max.1}$ characteristic of the interaction between the polar solvents and the PTFE fibers, a second maximum temperature $T_{Max.(X)}$, an intrinsic characteristic of the dispersive surface tension of the polar molecules, and a third geometric parameter $a_{Xmin.}$, proper to the molecule $X$ itself. The general expression of the surface area $a_X(T)$ of the polar molecules was given below:

$$a_X(T) = a_{Xmin.} \times \frac{(T_{Max.1} - T)}{(563.02 - T)^{1/2} \left(T_{Max.(X)} - T\right)^{1/2}} \tag{15}$$

The large effect of the temperature on surface areas of molecules was here highlighted.

On the other hand, we highlighted an important effect of the temperature on the standard enthalpy of the vaporization $\Delta H_{vap.}^0$ of the organic molecules' supposed constant by Chehimi et al. [59], and therefore proposed the following relation that can be used to

determine the specific contribution of the Gibbs free energy of the adsorption of polar molecules on alumina particles:

$$\Delta G_a^0(T) = \delta \Delta H_{vap.}^0(T) + \varepsilon \tag{16}$$

where $\delta$ and $\varepsilon$ are two constants of interaction.

*2.3. The New Lewis's Acid Base Parameters*

In several cases, the Gutmann method cannot be applied because the linearity of Equation (11) is not satisfied for many solid substrates. This classical relationship was corrected and a new equation was proposed by Hamieh et al. [77,81], by adding a third parameter, $K$, reflecting the amphoteric character of the oxide or polymer according to:

$$-\Delta H^{Sp} = K_A \times DN + K_D \times AN - K \times DN \times AN \tag{17}$$

By dividing by $AN$, we obtain:

$$\frac{-\Delta H^{Sp}}{AN} = K_A \frac{DN}{AN} + K_D - K \times DN \tag{18}$$

or:

$$X1 = K_D + K_A\,X_2 - K\,X_3 \tag{19}$$

with:

$$X_1 = -\frac{\Delta H^{sp}}{AN}, X_2 = \frac{DN}{AN}, \ X_3 = DN \text{ and } K = K(K_A K_D) \tag{20}$$

where $X_1$, $X_2$ and $X_3$ are known for every polar molecule, whereas $K_D$, $K_A$ and $K$ are the unknown parameters. The problem, given by Equation (16), is represented by a linear system for the N solvents and admits a unique solution for $N \geq 3$, giving the three unknown constants numbers: $K_D$, $K_A$ and $K$.

On the other hand, in a previous study, Hamieh [82] proposed new entropic acidic $\omega_A$ and basic $\omega_D$ parameters to determine the entropic acid base character of the solid surfaces by writing:

$$\left(-\Delta S_a^{sp}\right) = \omega_A\,DN' + \omega_D\,AN' \tag{21}$$

or

$$\left(\frac{-\Delta S_a^{sp}}{AN'}\right) = \omega_A\left(\frac{DN'}{AN'}\right) + \omega_D \tag{22}$$

## 3. Materials and Solvents

The different solid particles used in this study were furnished by Aldrich. Classical organic probes, characterized by their donor and acceptor numbers, were used in this study. The corrected acceptor number, $AN' = AN - AN^d$, was given by Riddle and Fowkes [83], who subtracted the contribution of the Van der Waals interactions (or dispersion forces). This acceptor number was normalized by Hamieh et al. [77,81], who proposed to use a dimensionless donor number $DN'$ and a dimensionless acceptor number $AN'$. All probes (Aldrich) were highly pure grade (i.e., 99%). The probes used were n-alkanes (pentane, hexane, heptane, octane and nonane); amphoteric solvents: acetonitrile, acetone; basic solvents: ethyl acetate, tetrahydrofuran (THF) and acidic solvents: chloroform and nitromethane.

In Table 2, we gave the donor and acceptor numbers of the polar probes used in this study. Chromatographic injections were performed using Hamilton microsyringes from Sigma-Aldrich, France. The measurements were carried out with a Focus GC Chromatograph equipped with a flame ionization detector of high sensitivity. The IGC measurements were performed on a commercial Focus GC gas chromatograph equipped with a flame ionization detector. Dried nitrogen was the carrier gas. The gas flow rate was set at

25 mL/min. The injector and detector temperatures were maintained at 400 K throughout the experiments.

**Table 2.** Normalized donor and acceptor numbers of polar molecules [62,64,79].

| Probes | *DN'* | *AN'* | *DN'/AN'* | Acid Base Force |
|---|---|---|---|---|
| CCl$_4$ | 0 | 2.3 | 0 | Acid |
| CHCl$_3$ | 0 | 18.7 | 0 | Stronger acidity |
| CH$_2$Cl$_2$ | 3 | 13.5 | 0.2 | Weaker amphoteric |
| Toluene | 9.75 | 3.3 | 3.0 | Amphoteric |
| Diethyl ether | 48 | 4.9 | 9.8 | Amphoteric |
| THF | 50 | 1.9 | 26.3 | Stronger Basicity |

To achieve an infinite dilution, 0.1 μL of each probe was injected with 1 μL Hamilton syringes taken from the vapor above the liquid solvent surface and emptied three times into air, in order to approach linear condition gas chromatography. The column temperatures were 300 K to 460 K, varied in 20 °C steps. Each probe injection was repeated three times, and the average retention time, $t_R$, was used for the calculation. The standard deviation was less than 1% in all measurements. All columns used in this study were prepared using a stainless-steel column with a 2 mm inner diameter and with an approximate length of 20 cm.

## 4. Results

### 4.1. Determination of the Gibbs Free Energy of Adsorption

The results of the IGC technique at an infinite dilution gave the net retention volume $Vn$ of n-alkanes and polar molecules adsorbed on alumina at various temperatures in the interval [323.15 K, 463.15 K]. This allowed the obtaining of the Gibbs free energy $\Delta G_a^0$ of adsorption by using the following fundamental thermodynamic relation of IGC:

$$\Delta G_a^0 = -RT \ ln \ V_n + \beta(T) \tag{23}$$

and $\beta(T)$ is given by:

$$\beta(T) = RT \ ln \left( \frac{sm\pi_0}{P_0} \right) \tag{24}$$

where $s$ is the specific surface area of alumina, and $m$ the mass of the solid particles introduced in the column. Additionally, $P_0$ and $\pi_0$ are, respectively, given by the Kemball and Rideal reference state [21] at $T_0 = 0 \ ^\circ C$ by:

$$P_0 = 1.013 \times 10^5 Pa \text{ and } \pi_0 = 6.08 \times 10^{-5} \ N \ m^{-1} \tag{25}$$

The obtained results are presented in Table 3.

**Table 3.** Variations of the Gibbs free energy $(-\Delta G_a^0$ in J mol$^{-1})$ of adsorption of the various polar solvents on alumina particles as a function of the temperature.

| T(K) | 303.15 | 323.15 | 343.15 | 363.15 | 383.15 | 403.15 | 423.15 | 443.15 | 463.15 |
|---|---|---|---|---|---|---|---|---|---|
| Pentane | 25,573 | 25,539 | 25,470 | 25,441 | 25,397 | 25,353 | 25,309 | 25,265 | 25,573 |
| Hexane | 28,968 | 28,878 | 28,790 | 28,698 | 28,603 | 28,522 | 28,428 | 28,338 | 28,968 |
| Heptane | 31,940 | 31,857 | 31,774 | 31,692 | 31,609 | 31,527 | 31,444 | 31,361 | 31,123 |
| Octane | 35,420 | 35,117 | 34,813 | 34,510 | 34,207 | 33,904 | 33,601 | 33,604 | 32,995 |
| Nonane | 38,821 | 38,467 | 37,716 | 37,163 | 36,611 | 36,058 | 35,506 | 34,953 | 34,401 |

**Table 3.** *Cont*.

| T(K) | 303.15 | 323.15 | 343.15 | 363.15 | 383.15 | 403.15 | 423.15 | 443.15 | 463.15 |
|---|---|---|---|---|---|---|---|---|---|
| $CH_2Cl_2$ | 61,952 | 59,637 | 57,919 | 56,367 | 54,442 | 52,966 | 51,248 | 49,509 | 47,769 |
| $CHCl_3$ | 45,147 | 42,524 | 40,448 | 38,512 | 36,838 | 34,950 | 32,911 | 31,850 | 29,664 |
| $CCl_4$ | 34,479 | 34,514 | 34,449 | 34,435 | 34,420 | 34,405 | 34,391 | 34,376 | 34,361 |
| THF | 64,519 | 62,228 | 60,464 | 58,838 | 57,449 | 55,865 | 54,281 | 53,324 | 51,507 |
| Ether | 67,319 | 65,062 | 63,377 | 61,763 | 60,317 | 58,729 | 56,976 | 55,555 | 53,967 |
| Toluene | 47,084 | 46,302 | 45,020 | 44,028 | 43,511 | 42,617 | 41,724 | 40,831 | 39,937 |

Table 3 clearly showed that the standard free energy of adsorption $\left(-\Delta G_a^0\right)$ decreases for every probe when the temperature increases (Figure 1) and decreasing at the same time the adsorption of the molecules. One also observed that the values of the free energy of adsorption of polar solvents are greater than that of n-alkanes, showing the strong polar interaction between the alumina and polar molecules and proving the importance of the Lewis's acid base character of the solid surfaces.

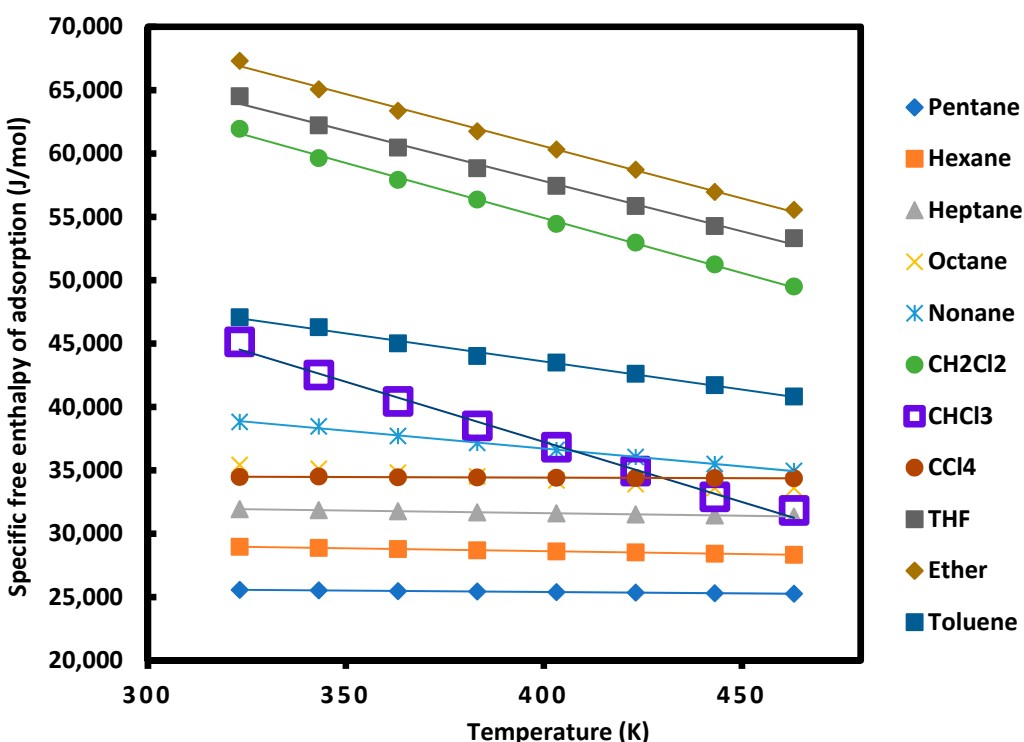

**Figure 1.** Variations of the surface free energy $\left(-\Delta G_a^0(T)\right.$ (*in* J/mol) of the solvents adsorbed on alumina particles as a function of the temperature *T* (K).

From Table 3 and Figure 1, it was deduced the values of the standard enthalpy and entropy of the adsorption for all organic molecules as shown in Table 4.

**Table 4.** Values of standard enthalpy $\left(-\Delta H_a^0 \left( \text{J mol}^{-1} \right)\right)$ and entropy $\left(-\Delta S_a^0 \left( \text{J K}^{-1} \text{ mol}^{-1} \right)\right)$ of adsorption of the various organic molecules adsorbed on alumina surfaces.

| Probes | $-\Delta H_a^0$ | $-\Delta S_a^0$ | Equation of $-\Delta G_a^0$ (T) | $R^2$ |
|---|---|---|---|---|
| Pentane | 26,284 | 2.2 | $-\Delta G_a^0 (T) = -2.2\,T + 26{,}284$ | 0.9967 |
| Hexane | 30,423 | 4.5 | $-\Delta G_a^0 (T) = -4.5\,T + 30{,}423$ | 0.9967 |

**Table 4.** *Cont.*

| Probes | $-\Delta H_a^0$ | $-\Delta S_a^0$ | Equation of $-\Delta G_a^0$ (T) | $R^2$ |
|---|---|---|---|---|
| Heptane | 33,192 | 4.1 | $-\Delta G_a^0 (T) = -4.1\,T + 33{,}192$ | 1.0000 |
| Octane | 40,094 | 15.4 | $-\Delta G_a^0 (T) = -15.4\,T + 40{,}094$ | 0.9989 |
| Nonane | 47,409 | 28.1 | $-\Delta G_a^0 (T) = -28.1\,T + 47{,}409$ | 0.9985 |
| CCl$_4$ | 34,696 | 0.7 | $-\Delta G_a^0 (T) = -0.7\,T + 34{,}696$ | 0.9991 |
| CHCl$_3$ | 72,931 | 93.8 | $-\Delta G_a^0 (T) = -93.8\,T + 72{,}931$ | 0.9949 |
| CH$_2$Cl$_2$ | 87,807 | 86.6 | $-\Delta G_a^0 (T) = -86.6\,T + 87{,}807$ | 0.9985 |
| Toluene | 60,493 | 44.4 | $-\Delta G_a^0 (T) = -44.4\,T + 60{,}493$ | 0.9978 |
| THF | 86,959 | 76.8 | $-\Delta G_a^0 (T) = -76.8\,T + 86{,}959$ | 0.9970 |
| Ether | 91,555 | 81.5 | $-\Delta G_a^0 (T) = -81.5\,T + 91{,}555$ | 0.9976 |

The polar molecules on Table 4 can be classified in decreasing order of standard enthalpy of adsorption:

$$\text{Diethyl ether} > CH_2Cl_2 > THF > CHCl_3 > \text{Toluene} > CCl_4 \qquad (26)$$

Inequalities (21) showed that the alumina material exhibits an amphoteric character because the stronger adsorption is obtained with the amphoteric molecule, such as diethyl ether.

In order to quantify the Lewis acid base properties, relation (10) was used by using the values of $\left(-\Delta H_a^0\right)$ of polar molecules. A linear relation was obtained giving:

$$\frac{\left(-\Delta H_a^0\right)}{AN} = 1.112\,\frac{DN}{AN} + 3.045 \qquad (27)$$

and the values of the Lewis enthalpic acid base constants are, respectively, $K_A = 1.112\,\text{kJ/mol}$ and $K_D = 3.045\,\text{kJ/mol}$, whereas, those relative to the entropic constants are $\omega_A = 1.44\,\text{J K}^{-1}\text{mol}^{-1}$ and $\omega_D = 4.56\,\text{J K}^{-1}\text{mol}^{-1}$. These can be written in unitless as: $K_A = 0.664$, $K_D = 1.820$, $\omega_A = 8.6 \times 10^{-4}$ and $\omega_D = 2.7 \times 10^{-3}$. The respective ratios of the enthalpic and entropic bases on acid constants are therefore given by:

$$\frac{K_D}{K_A} = 2.74 \text{ and } \frac{\omega_D}{\omega_A} = 3.2 \qquad (28)$$

These results confirmed that the alumina surfaces have an amphoteric behavior with a stronger Lewis's base character 2.74 times greater than the acidic character. In this case, we proved that the Lewis and Bronsted concepts are very close, showing the amphoteric behavior in both aqueous medium and solid surfaces.

*4.2. London Dispersive Surface Energy of Alumina Particles*

In this section, we calculated the London dispersive component of the alumina particles by using relations (3), (4), (12) and (13) and applying the various models of Van der Waals (VDW), Redlich–Kwong (R-K), Kiselev, geometric, cylindrical or spherical models [75–79], Dorris–Gray [69] and the thermal model [71]. In Table 5, we gave the values of the London dispersive surface energy $\gamma_s^d$ (T) of the alumina particles as a function of the temperature for the various methods and models.

The results in Table 5 and Figure 2 show the linear decrease of when the temperature increases for all applied models and methods.

Three groups can be distinguished between the above models and methods:

1. The group, constituted by the Kiselev, cylindrical, VDW, geometric and Doris–Gray models, taking into account the geometric form of n-alkanes, and they presented very close values of $\gamma_s^d$ and the surface of methylene group (Figure 3).

**Table 5.** Values of the dispersive component of the surface energy $\gamma_s^d$ $(\text{mJ/m}^2)$ of alumina particles as a function of the temperature.

| | | | | $\gamma_s^d$ **(mJ/m$^2$) (Alumina)** | | | | |
|---|---|---|---|---|---|---|---|---|
| *T* (K) | 323.15 | 343.15 | 363.15 | 383.15 | 403.15 | 423.15 | 443.15 | 463.15 |
| Kiselev | 53.0 | 47.1 | 41.7 | 37.8 | 31.4 | 23.2 | 23.6 | 22.9 |
| Cylindrical | 52.6 | 47.3 | 42.4 | 39.2 | 33.2 | 25.2 | 17.0 | 16.4 |
| VDW | 54.4 | 48.1 | 42.3 | 38.2 | 31.4 | 22.9 | 22.1 | 18.1 |
| Geometric | 40.4 | 37.0 | 34.1 | 32.4 | 28.6 | 23.0 | 22.8 | 22.6 |
| Redlich–Kwong | 88.8 | 78.5 | 69.1 | 62.3 | 51.3 | 37.3 | 33.9 | 26.6 |
| Spherical | 148.8 | 127.9 | 109.1 | 95.0 | 74.7 | 51.4 | 48.5 | 37.5 |
| Hamieh | 80.6 | 69.3 | 59.2 | 51.6 | 40.9 | 21.2 | 20.4 | 18.1 |
| Dorris–Gray | 59.8 | 54.8 | 50.9 | 50.6 | 46.8 | 42.8 | 42.2 | 41.1 |
| Hamieh–Gray | 105.6 | 88.7 | 74.9 | 67.0 | 55.0 | 44.1 | 37.2 | 30.2 |
| Global average | 76.0 | 66.5 | 58.2 | 52.7 | 43.7 | 32.3 | 29.7 | 25.9 |

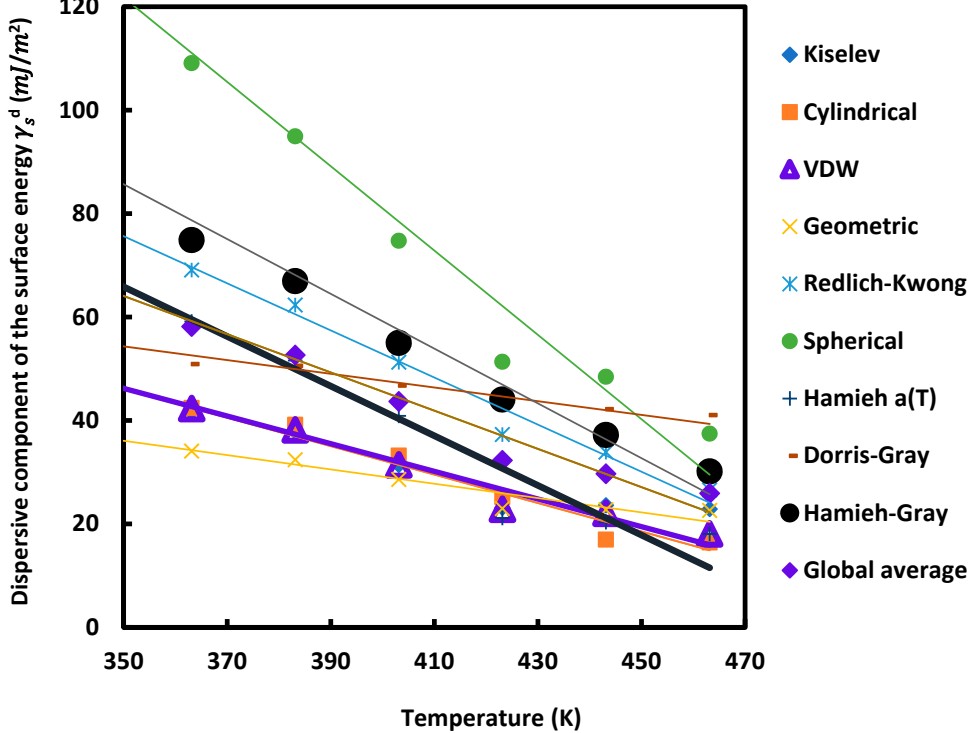

**Figure 2.** Dispersive component of the surface energy $\gamma_s^d$ $(\text{mJ/m}^2)$ of alumina particles as a function of the temperature *T* (K) using different methods and models.

2. The second group concerns the models relative to thermal model, Redlich–Kwong equation and the global average results that concluded to the more accurate values of the $\gamma_s^d$ of the alumina surfaces (Figure 3).
3. The third group is relative to the spherical model and Hamieh–Gray- model [69]. The obtained values of $\gamma_s^d$ are higher than the true value, certainly because of the non-accurate values of the surface area of n-alkanes for these models (Figure 3).

Figure 2 also showed that the global average results are very close to those obtained by the thermal model.

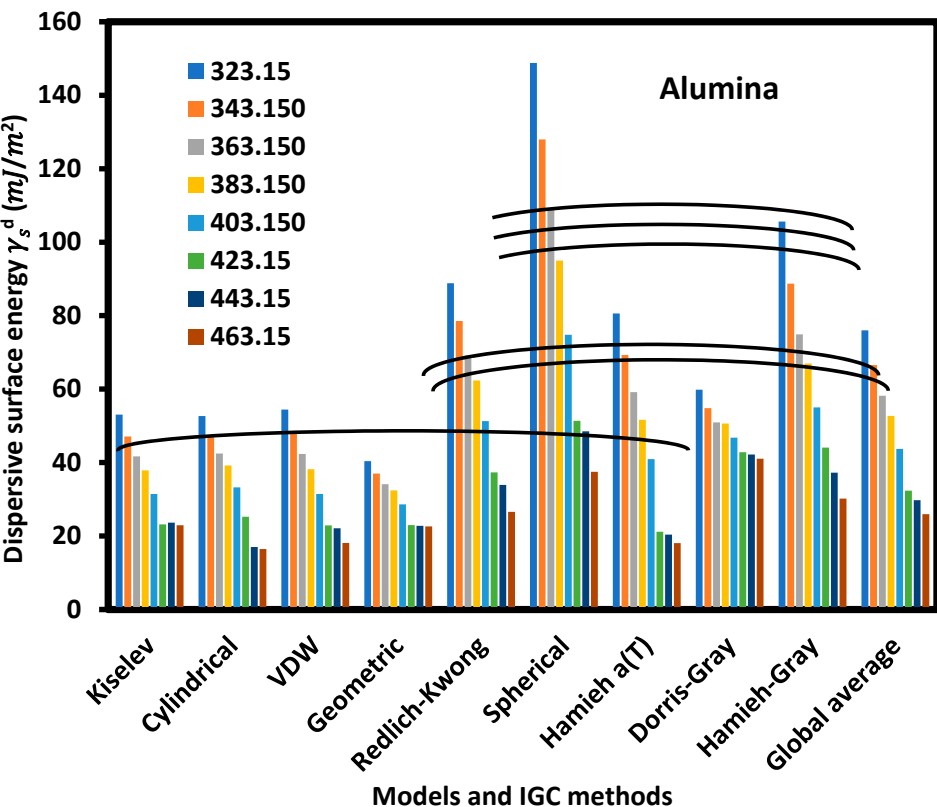

**Figure 3.** Dispersive surface energy $\gamma_s^d$ $(\mathrm{mJ/m^2})$ of alumina particles following the various methods and models for eight temperatures.

The different equations giving $\gamma_s^d(T)$ of alumina particles as a function of the temperature for the various molecular models of n-alkanes were presented in Table 6 with the values of the dispersive surface entropy $\varepsilon_s^d$, the extrapolated values $\gamma_s^d(T = 0K)$ and the maximum of temperature $T_{Max}$ allowed by the chosen molecular model. $T_{Max}$ was defined using the following relation:

$$T_{Max} = -\frac{\gamma_s^d(T = 0K)}{\varepsilon_s^d} \tag{29}$$

Table 6 showed the large difference between the values of the dispersive surface entropy of the different molecular models compared to those given by the Hamieh thermal model ($\varepsilon_s^d = -0.480$ mJ m$^{-2}$K$^{-1}$) and the highest value obtained by the spherical model. The important result obtained here is the average value of $T_{Max} = 523.42$ K, with comparable values to the various models, excepted for the case of Dorris–Gray.

**Table 6.** Equations $\gamma_s^d(T)$ of alumina particles for various molecular models of n-alkanes, the dispersive surface entropy $\varepsilon_s^d$, the extrapolated values $\gamma_s^d(T = 0K)$ and the maximum of temperature $T_{Max}$.

| Molecular Model | $\gamma_s^d(T)$ (mJ/m$^2$) | $\varepsilon_s^d = d\gamma_s^d/dT$ (mJ m$^{-2}$ K$^{-1}$) | $\gamma_s^d(T=0K)$ (mJ/m$^2$) | $T_{Max}(K)$ |
|---|---|---|---|---|
| Kiselev | $\gamma_s^d(T) = -0.232\,T + 126.4$ | $-0.232$ | 126.4 | 544.36 |
| Cylindrical | $\gamma_s^d(T) = -0.275\,T + 142.3$ | $-0.275$ | 142.3 | 517.41 |
| VDW | $\gamma_s^d(T) = -0.2674\,T + 139.8$ | $-0.267$ | 139.8 | 522.89 |
| Geometric | $\gamma_s^d(T) = -0.138\,T + 84.6$ | $-0.139$ | 84.6 | 610.58 |
| Redlich–Kwong | $\gamma_s^d(T) = -0.455\,T + 235.1$ | $-0.456$ | 235.1 | 516.05 |
| Spherical | $\gamma_s^d(T) = -0.815\,T + 407.2$ | $-0.815$ | 407.2 | 499.39 |

**Table 6.** *Cont.*

| Molecular Model | $\gamma_s^d(T)$ (mJ/m$^2$) | $\varepsilon_s^d = d\gamma_s^d/dT$ (mJ m$^{-2}$ K$^{-1}$) | $\gamma_s^d(T=0K)$ (mJ/m$^2$) | $T_{Max}(K)$ |
|---|---|---|---|---|
| Hamieh model | $\gamma_s^d(T) = 0.480\,T + 233.9$ | $-0.480$ | 233.9 | 487.21 |
| Dorris–Gray | $\gamma_s^d(T) = -0.132\,T + 100.7$ | $-0.133$ | 100.7 | 760.08 |
| Hamieh–Gray | $\gamma_s^d(T) = -0.500\,T + 271.0$ | $-0.530$ | 271.0 | 511.78 |
| Global average | $\gamma_s^d(T) = -0.370\,T + 141.2$ | $-0.370$ | 193.4 | 523.42 |

*4.3. Surface Thermodynamic of Alumina Particles*

4.3.1. The Gibbs Specific Free Energy of Adsorption

The variations of the Gibbs specific free energy ($\Delta G_a^{sp}(T)$) of the various polar solvents adsorbed on silica particle surface as a function of the temperature $T$ were given in Table S1 (Supplementary Materials) for the various models, such as the Kiselev, Van der Waals, Redlich–Kwong, geometric, spherical and thermal models and IGC methods, such as the boiling point, vapor pressure, deformation polarizability, enthalpy of vaporization, $\Delta H_{vap.}^0$, $\Delta H_{vap.}^0(T)$ and topological index methods.

The values $\Delta G_a^{sp}(T)$ presented on Table S1 proved that there is no universal method that can give accurate results, except the thermal model, which accounted for the temperature effect on the surface area of molecules. We observed, in Table S1, the large difference in results obtained with the different models and methods that can reach 100% of the results given by the thermal model. We showed, in Figure 4, the disparity in the obtained results with different polar molecules adsorbed on alumina surfaces. The linear variations of $\Delta G_a^{sp}(T)$, plotted in Figure 4 and Table 7 and giving the corresponding equations, clearly proved that the slope and the ordinate at the origin strongly depend on the chosen model or used method for every polar molecule.

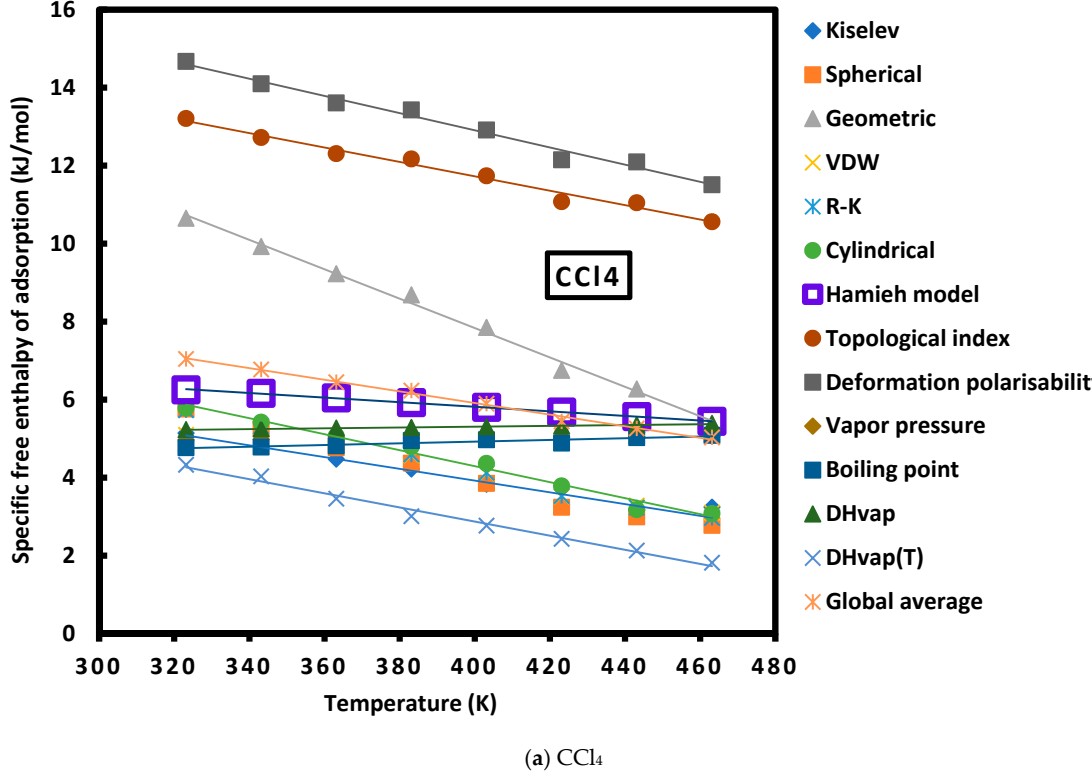

(**a**) CCl₄

**Figure 4.** *Cont.*

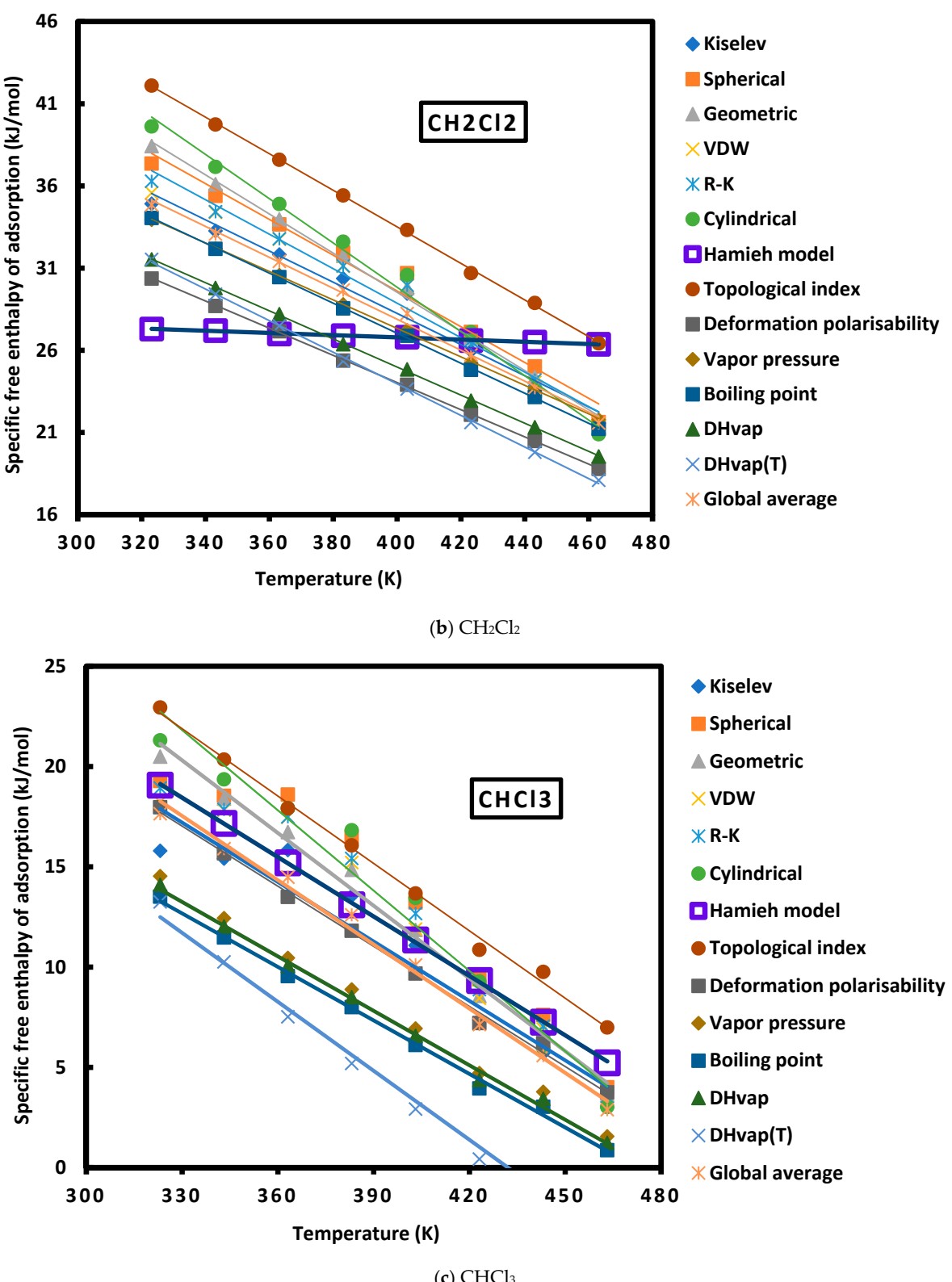

(**b**) CH₂Cl₂

(**c**) CHCl₃

**Figure 4.** *Cont.*

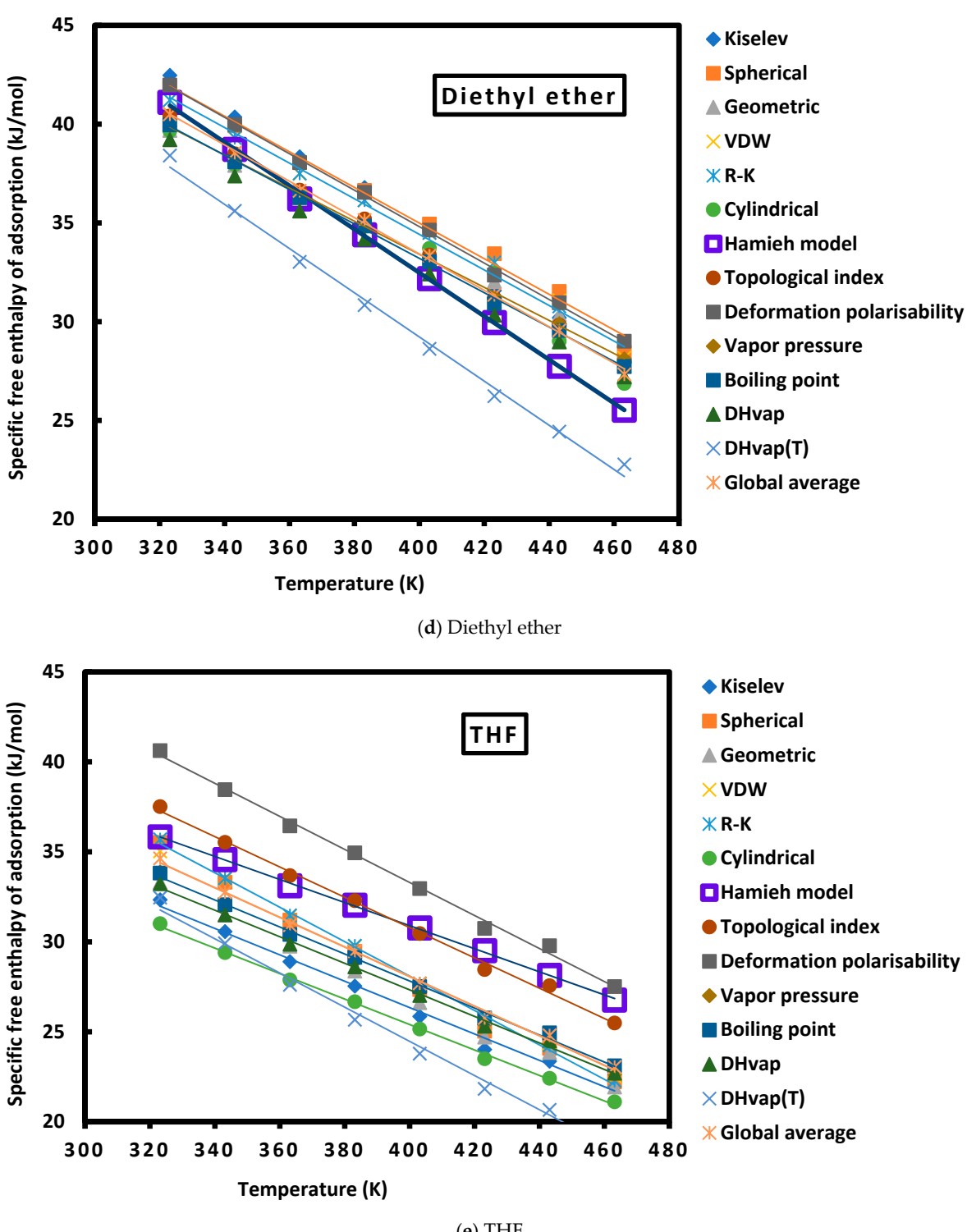

(**d**) Diethyl ether

(**e**) THF

**Figure 4.** *Cont.*

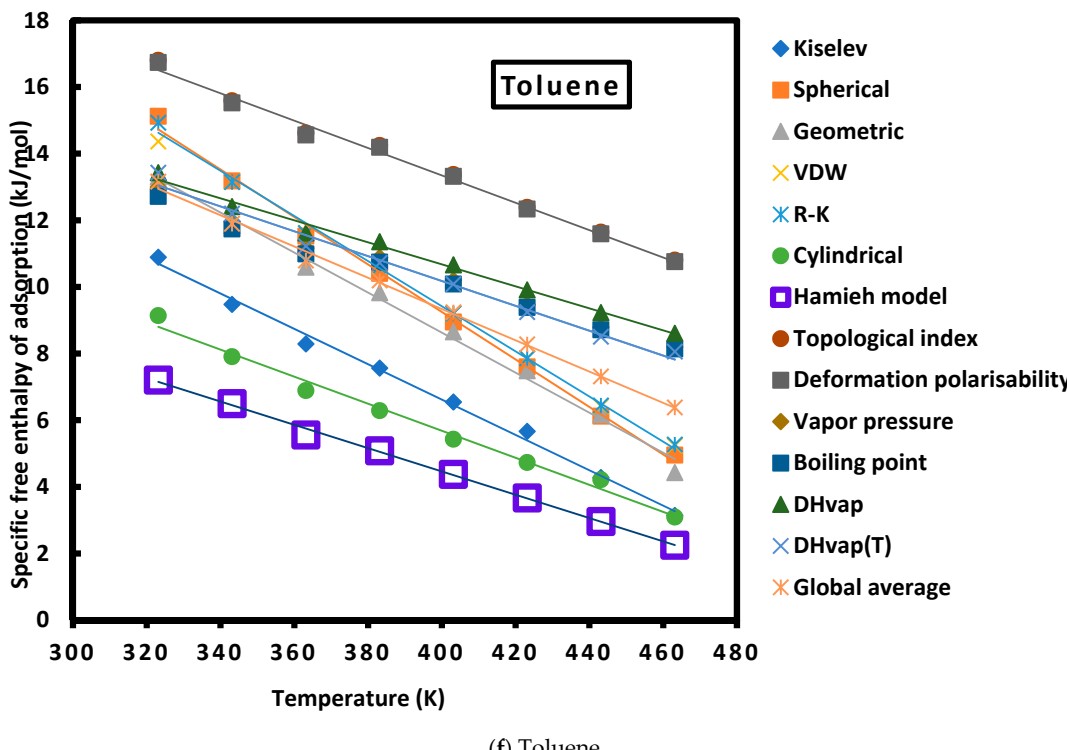

(**f**) Toluene

**Figure 4.** Variations of $\Delta G_a^{sp}$ as a function of the temperature for the various polar molecules adsorbed on the alumina surface by using the different IGC models and methods.

**Table 7.** The linear equations of $-\Delta G_a^{sp}(T)$ (kJ/mol) of the polar solvents adsorbed on alumina particles as a function of the temperature $T$ (K) for all models and methods.

| Model or Method | Polar Solvent | Equation $-\Delta G_a^{sp}(T)$ (kJ/mol) |
|---|---|---|
| Kiselev | $CCl_4$ | $-\Delta G_a^{sp}(T) = -0.015\,T + 9.951$ |
| | $CHCl_3$ | $-\Delta G_a^{sp}(T) = -0.0950\,T + 66.196$ |
| | $CH_2Cl_2$ | $-\Delta G_a^{sp}(T) = -0.099\,T + 49.816$ |
| | Diethyl ether | $-\Delta G_a^{sp}(T) = -0.104\,T + 76.237$ |
| | THF | $-\Delta G_a^{sp}(T) = -0.073\,T + 55.663$ |
| | Toluene | $-\Delta G_a^{sp}(T) = -0.053\,T + 27.836$ |
| Spherical | $CCl_4$ | $-\Delta G_a^{sp}(T) = -0.022\,T + 12.846$ |
| | $CHCl_3$ | $-\Delta G_a^{sp}(T) = -0.109\,T + 73.138$ |
| | $CH_2Cl_2$ | $-\Delta G_a^{sp}(T) = -0.115\,T + 58.418$ |
| | Diethyl ether | $-\Delta G_a^{sp}(T) = -0.090\,T + 71.08$ |
| | THF | $-\Delta G_a^{sp}(T) = -0.095\,T + 65.952$ |
| | Toluene | $-\Delta G_a^{sp}(T) = -0.071\,T + 37.748$ |
| Geometric | $CCl_4$ | $-\Delta G_a^{sp}(T) = -0.038\,T + 22.904$ |
| | $CHCl_3$ | $-\Delta G_a^{sp}(T) = -0.119\,T + 77.161$ |
| | $CH_2Cl_2$ | $-\Delta G_a^{sp}(T) = -0.121\,T + 60.18$ |
| | Diethyl ether | $-\Delta G_a^{sp}(T) = -0.081\,T + 65.801$ |
| | THF | $-\Delta G_a^{sp}(T) = -0.080\,T + 59.029$ |
| | Toluene | $-\Delta G_a^{sp}(T) = -0.060\,T + 32.654$ |

**Table 7.** *Cont*.

| Model or Method | Polar Solvent | Equation $-\Delta G_a^{sp}(T)$ (kJ/mol) |
|---|---|---|
| Van der Waals (VDW) | CCl$_4$ | $-\Delta G_a^{sp}(T) = -0.017\,T + 10.919$ |
| | CHCl$_3$ | $-\Delta G_a^{sp}(T) = -0.101\,T + 69.275$ |
| | CH$_2$Cl$_2$ | $-\Delta G_a^{sp}(T) = -0.111\,T + 56.128$ |
| | Diethyl ether | $-\Delta G_a^{sp}(T) = -0.085\,T + 68.404$ |
| | THF | $-\Delta G_a^{sp}(T) = -0.091\,T + 64.627$ |
| | Toluene | $-\Delta G_a^{sp}(T) = -0.066\,T + 35.609$ |
| Redlich–Kwong (R-K) | CCl$_4$ | $-\Delta G_a^{sp}(T) = -0.021\,T + 12.349$ |
| | CHCl$_3$ | $-\Delta G_a^{sp}(T) = -0.105\,T + 70.824$ |
| | CH$_2$Cl$_2$ | $-\Delta G_a^{sp}(T) = -0.113\,T + 57.257$ |
| | Diethyl ether | $-\Delta G_a^{sp}(T) = -0.090\,T + 70.46$ |
| | THF | $-\Delta G_a^{sp}(T) = -0.096\,T + 66.356$ |
| | Toluene | $-\Delta G_a^{sp}(T) = -0.068\,T + 36.511$ |
| Cylindrical | CCl$_4$ | $-\Delta G_a^{sp}(T) = -0.021\,T + 12.489$ |
| | CHCl$_3$ | $-\Delta G_a^{sp}(T) = -0.135\,T + 83.700$ |
| | CH$_2$Cl$_2$ | $-\Delta G_a^{sp}(T) = -0.136\,T + 65.871$ |
| | Diethyl ether | $-\Delta G_a^{sp}(T) = -0.088\,T + 68.367$ |
| | THF | $-\Delta G_a^{sp}(T) = -0.071\,T + 53.71$ |
| | Toluene | $-\Delta G_a^{sp}(T) = -0.041\,T + 21.91$ |
| Hamieh model | CCl$_4$ | $-\Delta G_a^{sp}(T) = -0.006\,T + 8.164$ |
| | CH$_2$Cl$_2$ | $-\Delta G_a^{sp}(T) = -0.007\,T + 29.475$ |
| | CHCl$_3$ | $-\Delta G_a^{sp}(T) = -0.099\,T + 51.024$ |
| | Diethyl ether | $-\Delta G_a^{sp}(T) = -0.110\,T + 76.509$ |
| | THF | $-\Delta G_a^{sp}(T) = -0.064\,T + 56.551$ |
| | Toluene | $-\Delta G_a^{sp}(T) = -0.035\,T + 18.456$ |
| Topological index | CCl$_4$ | $-\Delta G_a^{sp}(T) = -0.019\,T + 19.115$ |
| | CH$_2$Cl$_2$ | $-\Delta G_a^{sp}(T) = -0.111\,T + 77.995$ |
| | CHCl$_3$ | $-\Delta G_a^{sp}(T) = -0.112\,T + 58.858$ |
| | Diethyl ether | $-\Delta G_a^{sp}(T) = -0.088\,T + 68.894$ |
| | THF | $-\Delta G_a^{sp}(T) = -0.084\,T + 64.482$ |
| | Toluene | $-\Delta G_a^{sp}(T) = -0.041\,T + 29.895$ |
| Deformation polarizability | CCl$_4$ | $-\Delta G_a^{sp}(T) = -0.022\,T + 21.723$ |
| | CH$_2$Cl$_2$ | $-\Delta G_a^{sp}(T) = -0.083\,T + 57.101$ |
| | CHCl$_3$ | $-\Delta G_a^{sp}(T) = -0.100\,T + 50.004$ |
| | Diethyl ether | $-\Delta G_a^{sp}(T) = -0.0922\,T + 71.692$ |
| | THF | $-\Delta G_a^{sp}(T) = -0.092\,T + 70.019$ |
| | Toluene | $-\Delta G_a^{sp}(T) = -0.041\,T + 29.774$ |

**Table 7.** *Cont.*

| Model or Method | Polar Solvent | *Equation* $-\Delta G_a^{sp}(T)$ (kJ/mol) |
|---|---|---|
| Vapor pressure | CCl$_4$ | $-\Delta G_a^{sp}(T) = 0.001\ T + 4.7609$ |
| | CH$_2$Cl$_2$ | $-\Delta G_a^{sp}(T) = -0.087\ T + 61.958$ |
| | CHCl$_3$ | $-\Delta G_a^{sp}(T) = -0.091\ T + 43.784$ |
| | Diethyl ether | $-\Delta G_a^{sp}(T) = -0.084\ T + 66.903$ |
| | THF | $-\Delta G_a^{sp}(T) = -0.079\ T + 59.071$ |
| | Toluene | $-\Delta G_a^{sp}(T) = -0.033\ T + 23.369$ |
| Boiling point | CCl$_4$ | $-\Delta G_a^{sp}(T) = 0.002\ T + 4.0546$ |
| | CH$_2$Cl$_2$ | $-\Delta G_a^{sp}(T) = -0.091\ T + 63.571$ |
| | CHCl$_3$ | $-\Delta G_a^{sp}(T) = -0.089\ T + 42.024$ |
| | Diethyl ether | $-\Delta G_a^{sp}(T) = -0.087\ T + 68.002$ |
| | THF | $-\Delta G_a^{sp}(T) = -0.075\ T + 57.849$ |
| | Toluene | $-\Delta G_a^{sp}(T) = -0.031\ T + 22.645$ |
| Enthalpy of vaporization $\Delta H$vap(298K) | CCl$_4$ | $-\Delta G_a^{sp}(T) = 0.001\ T + 4.8875$ |
| | CH$_2$Cl$_2$ | $-\Delta G_a^{sp}(T) = -0.086\ T + 59.17$ |
| | CHCl$_3$ | $-\Delta G_a^{sp}(T) = -0.091\ T + 43.106$ |
| | Diethyl ether | $-\Delta G_a^{sp}(T) = -0.086\ T + 66.757$ |
| | THF | $-\Delta G_a^{sp}(T) = -0.074\ T + 56.843$ |
| | Toluene | $-\Delta G_a^{sp}(T) = -0.033\ T + 23.885$ |
| Thermic enthalpy of vaporization $\Delta H$vap(T) | CCl$_4$ | $-\Delta G_a^{sp}(T) = -0.018\ T + 10.116$ |
| | CH$_2$Cl$_2$ | $-\Delta G_a^{sp}(T) = -0.096\ T + 62.393$ |
| | CHCl$_3$ | $-\Delta G_a^{sp}(T) = -0.115\ T + 49.546$ |
| | Diethyl ether | $-\Delta G_a^{sp}(T) = -0.112\ T + 73.958$ |
| | THF | $-\Delta G_a^{sp}(T) = -0.095\ T + 62.454$ |
| | Toluene | $-\Delta G_a^{sp}(T) = -0.037\ T + 25.095$ |

### 4.3.2. Lewis's Acid Base Parameters

From the equations of $\Delta G_a^{sp}(T)$ in Table 7, we deduced the values of $(-\Delta H_a^{sp})$ and $(-\Delta S_a^{sp})$ relative to the adsorption of CCl$_4$, CH2Cl2, CHCl$_3$, diethyl ether, THF and toluene on alumina particles, by using the various molecular models and methods. The results are presented on Tables 8 and 9.

**Table 8.** Values of the specific enthalpy $(-\Delta H_a^{sp}\ in\ \text{J K}^{-1}\text{mol}^{-1})$ of the various polar solvents adsorbed on alumina by using the various molecular models, Hamieh model, topological index, deformation polarizability and vapor pressure methods compared to the global average with the standard deviation and the error percentage.

| Probes | CCl$_4$ | CHCl$_3$ | CH$_2$Cl$_2$ | Diethyl Ether | THF | Toluene |
|---|---|---|---|---|---|---|
| Kiselev | 9.95 | 49.82 | 66.20 | 76.24 | 55.66 | 27.84 |
| Spherical | 12.85 | 58.42 | 73.14 | 71.08 | 65.95 | 37.75 |
| Geometric | 22.90 | 60.18 | 77.16 | 65.80 | 59.03 | 32.65 |
| VDW | 10.92 | 56.13 | 69.28 | 68.40 | 64.63 | 35.61 |

**Table 8.** *Cont.*

| Probes | CCl$_4$ | CHCl$_3$ | CH$_2$Cl$_2$ | Diethyl Ether | THF | Toluene |
|---|---|---|---|---|---|---|
| R-K | 12.35 | 57.26 | 70.82 | 70.46 | 66.36 | 36.51 |
| Cylindrical | 12.49 | 65.87 | 83.70 | 68.37 | 53.71 | 21.91 |
| Hamieh model | 8.16 | 51.02 | 29.48 | 76.51 | 56.55 | 18.46 |
| Topological index | 19.12 | 58.86 | 78.00 | 68.89 | 64.48 | 29.90 |
| Deformation polarizability | 21.72 | 50.00 | 57.10 | 71.69 | 70.02 | 29.77 |
| Vapor pressure | 4.76 | 43.78 | 61.96 | 66.90 | 59.07 | 23.37 |
| Boiling point | 4.05 | 42.02 | 63.57 | 68.00 | 57.85 | 22.65 |
| $\Delta Hvap$(298K) | 4.89 | 43.11 | 59.17 | 66.76 | 56.84 | 23.89 |
| $\Delta Hvap(T)$ | 10.12 | 49.55 | 62.39 | 73.96 | 62.45 | 25.10 |
| Average values | 11.87 | 52.77 | 65.54 | 70.24 | 60.97 | 28.11 |
| Standard deviation | 6.16 | 7.34 | 13.41 | 3.52 | 4.99 | 6.17 |
| Error percentage | 51.86 | 13.91 | 20.47 | 5.01 | 8.18 | 21.96 |

**Table 9.** Values of the specific entropy ($-\Delta S_a^{sp}$ *in* J K$^{-1}$mol$^{-1}$) of the various polar solvents adsorbed on alumina by using the various molecular models, Hamieh model, topological index, deformation polarizability and vapor pressure methods, compared to global average with the standard deviation and the error percentage.

| Probes | CCl$_4$ | CHCl$_3$ | CH$_2$Cl$_2$ | Diethyl Ether | THF | Toluene |
|---|---|---|---|---|---|---|
| Kiselev | 15.1 | 98.8 | 94.9 | 104.1 | 73.3 | 53 |
| Spherical | 22.2 | 114.5 | 108.8 | 90.2 | 95.2 | 71.3 |
| Geometric | 37.7 | 120.8 | 119.1 | 80.8 | 80.2 | 60.1 |
| VDW | 17 | 111.1 | 101.1 | 84.9 | 91.4 | 65.6 |
| R-K | 20.5 | 113.3 | 104.9 | 90.1 | 95.7 | 67.7 |
| Cylindrical | 20.5 | 133.5 | 134.7 | 87.7 | 70.8 | 40.6 |
| Hamieh model | 5.9 | 98.7 | 6.7 | 110.1 | 64.1 | 35 |
| Topological index | 18.5 | 112 | 111.2 | 88.4 | 84.2 | 41.2 |
| Deformation polarizability | 22 | 99.9 | 82.7 | 92.2 | 91.8 | 41.1 |
| Vapor pressure | −0.6 | 91.2 | 86.6 | 83.7 | 78.8 | 32.8 |
| Boiling point | −2.2 | 88.9 | 91.3 | 87 | 75.1 | 31.4 |
| $\Delta Hvap$(298K) | −1 | 90.5 | 85.5 | 85.5 | 73.8 | 33 |
| $\Delta Hvap(T)$ | 18.1 | 114.7 | 96.1 | 111.8 | 95 | 37.3 |
| Average values | 14.9 | 106.8 | 94.1 | 92.0 | 82.3 | 46.9 |
| Standard deviation | 11.48 | 13.31 | 30.08 | 10.07 | 10.68 | 14.61 |
| Error percentage | 77.02 | 12.46 | 31.97 | 10.94 | 12.98 | 31.15 |

The calculations of the average, standard deviation and error percentage committed on the values of the specific enthalpy of adsorption on alumina reflect the dispersion of the results obtained by the models and methods compared to the thermal model. One concluded here that the error with CCl$_4$ is 51.86%, followed by toluene (21.96%), CH$_2$Cl$_2$ (20.47%), CHCl$_3$ (13.91%), THF (8.18%) and diethyl ether (5.01%). The results in Table 8 showed that the IGC methods that better match the thermal models are the following:

boiling point, vapor pressure and enthalpy of vaporization; followed by the other molecular models, such as the cylindrical and Kiselev models.

For the specific entropy of adsorption (Table 9), the error percentage reaches 77.02% with CCl$_4$ and 31.97% with CH$_2$Cl$_2$, followed by THF (12.98%), CHCl$_3$ (12.46%), THF (8.18%) and diethyl ether (5.01%). The closer methods to the thermal models are identical to those obtained with the specific enthalpy of adsorption, proving the effect of the temperature on the surface area of the organic molecules.

The acid base parameters of alumina were obtained from Tables 8 and 9 and allowed to plot Figures 5 and 6, representing the respective variations of $\left(\frac{-\Delta H_a^{sp}}{AN'}\right)$ and $\left(\frac{-\Delta S_a^{sp}}{AN'}\right)$ as a function of $\left(\frac{DN'}{AN'}\right)$. The excellent linearity was obtained with the different models given in Table 10; the values of the Lewis enthalpic acid base constants $K_A$ and $K_D$ and the Lewis entropic acid base constants $\omega_A$ and $\omega_D$ of the alumina surface. The values of the acid base parameters, which obtained the thermal model, are in the following:

$$K_A = 0.624, \; K_D = 1.831 \text{ and } K_D/K_A = 2.93$$
$$\omega_A = 0.72, \; \omega_D = 2.79 \text{ and } \omega_D \,/\, \omega_A = 3.9$$

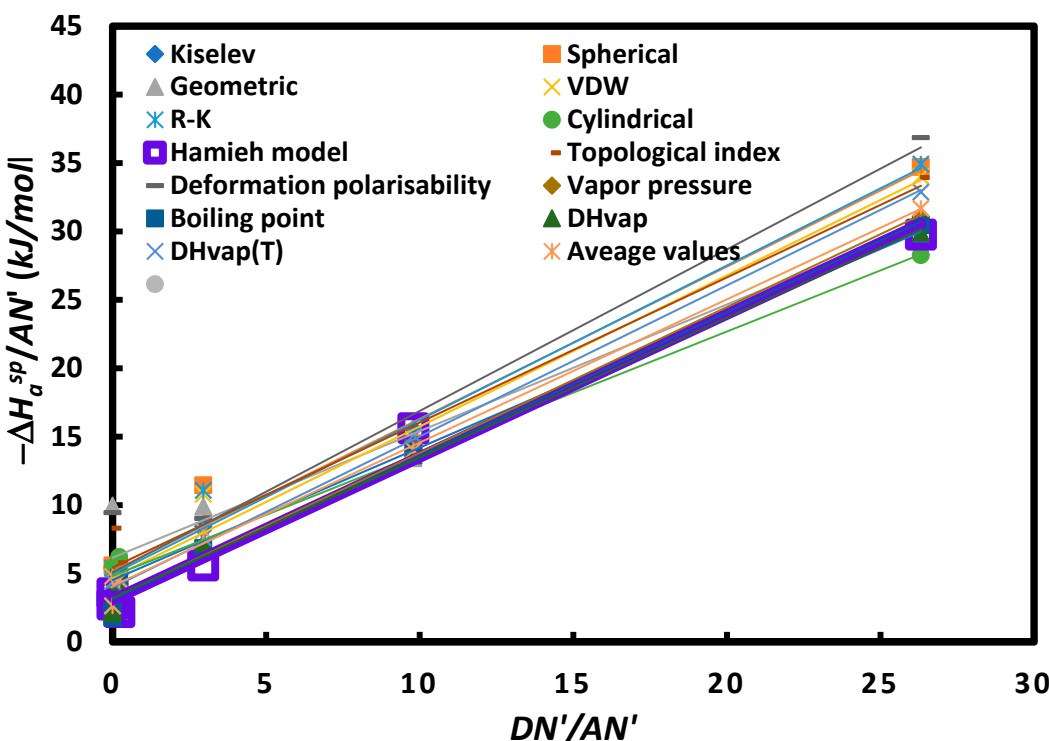

**Figure 5.** Variations of $\left(\frac{-\Delta H_a^{sp}}{AN'}\right)$ as a function of $\left(\frac{DN'}{AN'}\right)$ of different polar molecules adsorbed on alumina surface for different molecular models and IGC methods.

These results proved that the amphoteric character of the alumina surfaces with stronger Lewis's basicity are approximately three times more important than the Lewis's acidity of alumina. The same tendency was observed with the entropic acid base parameters.

Table 10 showed that the best models or methods that gave results comparable to those obtained by the thermal model are the following: boiling point, vapor pressure and enthalpy of vaporization methods, proving the important effect of the temperature on the surface areas adsorbed on alumina particles.

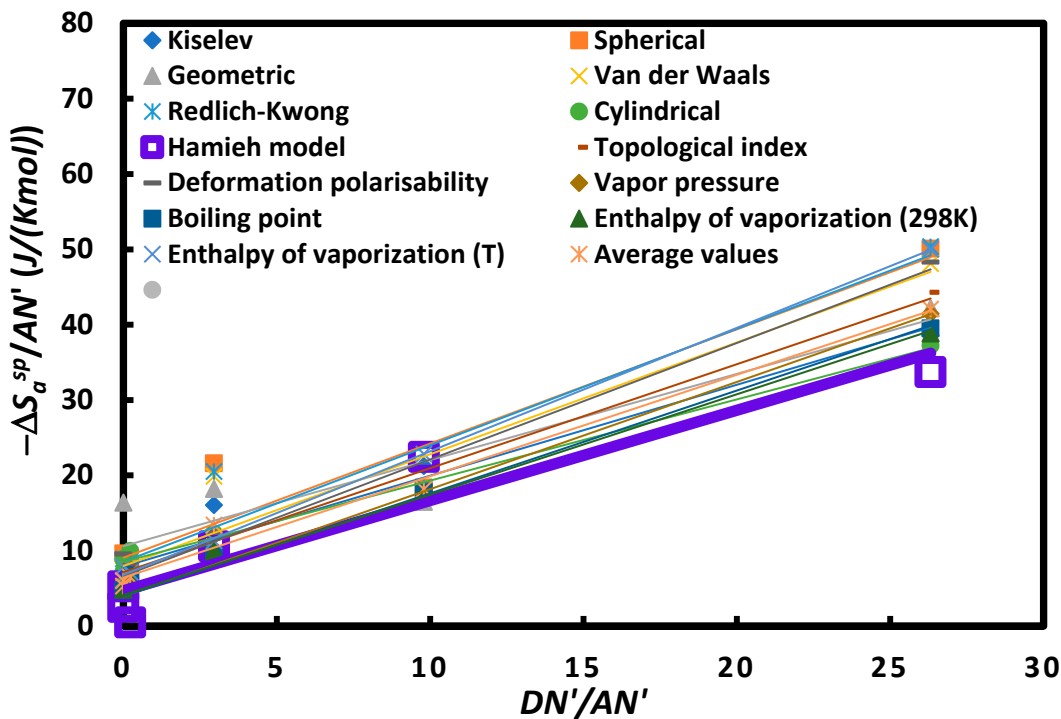

**Figure 6.** Variations of $\left(\frac{-\Delta S_a^{sp}}{AN'}\right)$ as a function of $\left(\frac{DN'}{AN'}\right)$ of different polar molecules adsorbed on alumina particles for different molecular models and IGC methods.

On the other hand, we observed similar results between the thermal model [69] and those given by Equations (27) and (28) by using the values of the standard enthalpy $(-\Delta H_a^0)$ and the entropy $(-\Delta S_a^0)$ of adsorption of polar molecules on alumina surfaces. The last way can resolve some difficulties related to the choice of the best model for more accurate results of the surface thermodynamic properties of materials.

To complete this section, it the specific statistical probability of the interaction $\Omega_a^{sp}$ between the polar probes and the solid surface was determined. The used relation is the following:

$$\Delta S_a^{sp} = R \, ln \, \Omega_a^{sp} \tag{30}$$

The obtained results are given in Table S2 for the various polar molecules and for the different used models and methods. The more interesting result, obtained here, concerned the value of the specific statistical probability of the interaction of $CCl_4$. Table S1 proved the presence of a maximum of $\Omega_a^{sp}$ for all models and IGC methods, followed by $CH_2Cl_2$ and toluene by using the Hamieh model. This result again proved the strong basicity of the alumina particles. Indeed, the strong probability of more acidic solvents is the highest compared to other solvents.

**Table 10.** Values of the enthalpic acid base constants, $K_A$ and $K_D$ (unitless), and the entropic acid base constants, $\omega_A$ and $\omega_D$ (unitless), of alumina surface and the acid base ratios for the different used molecular models and IGC methods.

| Models and IGC Methods | $K_A$ | $K_D$ | $K_D/K_A$ | $10^3 \cdot \omega_A$ | $10^3 \cdot \omega_D$ | $\omega_D / \omega_A$ |
|---|---|---|---|---|---|---|
| Kiselev | 0.578 | 2.705 | 4.68 | 0.72 | 4.71 | 6.5 |
| Spherical | 0.665 | 3.093 | 4.65 | 0.91 | 5.42 | 6.0 |
| Geometric | 0.553 | 3.676 | 6.65 | 0.68 | 6.34 | 9.3 |
| VDW | 0.659 | 2.818 | 4.28 | 0.89 | 4.76 | 5.4 |
| R-K | 0.674 | 2.961 | 4.40 | 0.92 | 5.11 | 5.5 |

**Table 10.** *Cont.*

| Models and IGC Methods | $K_A$ | $K_D$ | $K_D/K_A$ | $10^3 \cdot \omega_A$ | $10^3 \cdot \omega_D$ | $\omega_D / \omega_A$ |
|---|---|---|---|---|---|---|
| Cylindrical | 0.534 | 2.879 | 5.39 | 0.64 | 5.09 | 7.9 |
| Hamieh model | 0.624 | 1.831 | 2.93 | 0.72 | 2.79 | 3.9 |
| Topological index | 0.633 | 3.250 | 5.13 | 0.82 | 4.27 | 5.2 |
| Deformation polarizability | 0.705 | 3.034 | 4.30 | 0.92 | 3.97 | 4.3 |
| Vapor pressure | 0.637 | 1.887 | 2.96 | 0.85 | 2.35 | 2.8 |
| Boiling point | 0.626 | 1.863 | 2.97 | 0.82 | 2.36 | 2.9 |
| DHvap | 0.612 | 1.928 | 3.15 | 0.80 | 2.46 | 3.1 |
| DHvap(T) | 0.659 | 2.376 | 3.60 | 0.98 | 4.07 | 4.2 |
| Average values | 0.628 | 2.639 | 4.20 | 0.82 | 4.13 | 5.0 |
| Standard deviation | 0.05 | 0.61 | | 0.10 | 1.29 | |
| Error percentage | 7.78 | 22.96 | | 12.79 | 31.34 | |

## 5. Study of the Surface Properties of Other Oxides

### 5.1. Case of TiO$_2$ Particles

In a previous study [84], we determined that the surface properties of titanium dioxide particles constituted 80% anatase and 20% rutile; more particularly, the specific interactions of adsorption, the Lewis acid base and the surface energy of this catalyst by inverse gas chromatography (IGC) at infinite dilution.

We applied our new methodology to TiO$_2$ particles that exhibited a specific surface area of 59 m$^2$/g. We used the values of the surface areas of the organic probes given as a function of the temperature given by relations (13)–(15) to calculate the London dispersive surface energy and the Lewis acid–base parameters. The obtained results are presented in Table 11.

**Table 11.** London dispersive surface energy and enthalpic and entropic Lewis acid–base parameters of titania.

| Equation $\gamma_s^d$ (*T*) of TiO$_2$ (in mJ/m$^2$), *T* in K | $\gamma_s^d$ (*T*) = −0.484*T* + 231.5 |
|---|---|
| $K_A$ | 0.10 |
| $K_D$ | 0.97 |
| $K_D/$ | 9.72 |
| $\omega_A$ | $0.23 \times 10^{-3}$ |
| $\omega_D$ | $2.71 \times 10^{-3}$ |
| $\omega_D / \omega_A$ | 11.60 |

Table 11 obviously showed the stronger basicity of TiO$_2$ particles, which is ten times more basic than acidic and also proved a decrease of the London dispersive surface energy from 79.8 mJ/m$^2$ to 41.0 mJ/m$^2$ in the temperature interval [40 °C; 120 °C].

### 5.2. Case of SiO$_2$ Particles

We previously studied the thermodynamic surface properties of silica particles by using the IGC technique at infinite dilution and applying the thermal model [83]. The results were collected in Table 12.

**Table 12.** Thermodynamic surface parameters of silica particles.

| Equation $\gamma_s^d$ (*T*) of SiO$_2$ (in mJ/m$^2$), *T* in K | $\gamma_s^d$ (*T*) = −0.99*T* + 428 |
|---|---|
| $K_A$ | 0.23 |
| $K_D$ | 2.7 |
| $K_D/K_A$ | 11.60 |
| $\omega_A$ | $1.21 \times 10^{-3}$ |
| $\omega_D$ | $-1.38 \times 10^{-3}$ |
| $\omega_D / \omega_A$ | −1.14 |

The obtained results proved that the silica particles exhibited stronger Lewis acidic character, which were approximately 12 times more acidic than basic. We observed that $\gamma_s^d$ (*T*) decreased from 118.0 mJ/m$^2$ to 38.8 mJ/m$^2$ in the temperature interval [40 °C; 120 °C].

*5.3. Comparison between the Three Oxides*

To compare between alumina, titania and silica, we gave the corresponding results relative to the three oxides on Table 13.

**Table 13.** Surface characteristics of oxide particles.

| Parameter | Silica | Alumina | Titania |
|---|---|---|---|
| $\gamma_s^d$ (*T*) of xide | $\gamma_s^d$ (*T*) = −0.99*T* + 428 | $\gamma_s^d$(*T*) = 0.480T + 233.9 | $\gamma_s^d$ (*T*) = −0.484T + 231.5 |
| $K_A$ | 2.7 | 0.62 | 0.10 |
| $K_D$ | 0.23 | 1.83 | 0.97 |
| $K_D/K_A$ | 0.09 | 2.93 | 9.72 |
| $\omega_A$ | $1.21 \times 10^{-3}$ | $0.72 \times 10^{-3}$ | $0.23 \times 10^{-3}$ |
| $\omega_D$ | $-1.38 \times 10^{-3}$ | $2.79 \times 10^{-3}$ | $2.71 \times 10^{-3}$ |
| $\omega_D / \omega_A$ | −1.14 | 3.9 | 11.60 |

We can classify the above oxides in a decreasing order of London dispersive surface energy as follows:

$$\text{SiO}_2 > \text{Al}_2\text{O}_3 > \text{TiO}_2$$

On the other hand, the same order of these oxides is conserved in a decreasing order of acidity, we have:

$$\text{SiO}_2 > \text{Al}_2\text{O}_3 > \text{TiO}_2$$

For the ratios of $K_D/K_A$ and $\omega_D / \omega_A$, we observed an inversion of the previous order:

$$\text{TiO}_2 > \text{Al}_2\text{O}_3 > \text{SiO}_2$$

This means that when the global basicity increases, the London dispersive surface energy of materials decreases.

This interesting result confirmed previous results in the literature showing that $\gamma_s^d$ increases when the acidity of solid surface increases [85,86].

**6. Conclusions**

The surface thermodynamic properties, such as the London dispersive component of the surface energy, the Gibbs free energy of adsorption, the specific interactions and

the Lewis's acid base parameters of alumina, silica and titania particles, were determined by using the inverse gas chromatography technique at infinite dilution and applying 15 molecular models and chromatographic methods.

The determination of the London dispersive surface energy of alumina by using the various molecular models showed that Dorris–Gray formula [71] and Schultz et al. method [73] cannot be used for an accurate determination of $\gamma_s^d$ of solid alumina. We corrected the calculation of $\gamma_s^d(T)$ by using the new thermal model that considered the change of the surface areas $a(T)$ of organic molecules as a function of the temperature. We also applied our new methodology to determine the London dispersive surface energy of silica and titania particles. It was shown that the oxides were classified in decreasing order of their London dispersive surface energy:

$$SiO_2 > Al_2O_3 > TiO_2$$

The same order was conserved in the acidity of these solid materials and confirmed other results of the literature.

The specific Gibbs free energy was obtained by the different molecular models and methods. All methods were proved to be inaccurate except that of the thermal model. However, the methods based on the effect of the temperature on the thermodynamic parameters, such as the vapor pressure, the boiling point temperature and the standard enthalpy of vaporization, gave results closer to those obtained by the Hamieh thermal model [69]. We gave below the variations of the specific Gibbs free energy $\Delta G_a^{sp}(T)$ of the polar molecules adsorbed on the alumina surfaces as a function of the temperature with the values of the specific enthalpy and entropy of adsorption.

It was proved that the alumina surface is about three times more basic than acidic, the titania particles more strongly basic than acidic, whereas the silica surface was the more acidic material. The same tendency was observed with the entropic acid–base parameters. There were comparable values of the acid–base parameters of the different materials.

The new methodology proposed in this study will therefore allow the obtaining of an accurate determination of the London dispersive surface energy, the Gibbs free energy, the specific enthalpy and entropy of the adsorption of polar molecules on the solid surfaces, as well as the Lewis acid base parameters of solid substrates.

**Supplementary Materials:** The following supporting information can be downloaded at: https://www.mdpi.com/article/10.3390/appliedchem3020015/s1, Table S1: Values (in kJ/mol) of the specific free energy $(-\Delta G_a^{sp}(T))$ of the various polar solvents adsorbed on alumina particles surface for different temperatures by using the various molecular models and IGC methods.

**Funding:** This research received no external funding.

**Data Availability Statement:** Support information.

**Conflicts of Interest:** The author declares no conflict of interest.

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
