# Peer review of "New Physicochemical Methodology for the Determination of the Surface Thermodynamic Properties of Solid Particles"

_appliedchem, doi:10.3390/appliedchem3020015_

Round 1

Reviewer 1 Report

In this manuscript, for detecting the physicochemical properties of the particle surface, the authors propose a new method and validate it using alumina. However, the authors still need to make major revises and explanations. Some comments are as follows:

1. In the paper, the authors propose a new method to verify accuracy using only alumina and there is no study for other metal or non-metal particles. But the title is “New physicochemical methodology for the determination of the surface thermodynamic properties of solid particles”. We are skeptical about the suitability of other metals or non-metals. Please describe in detail the reliability of the method.

2. The 2.1 “Classical methods” of the manuscript is summarized the previous work, which is not necessary in a research-based paper. These can be put in the introduction, and the manuscript needs to describe in more detail the new approach proposed by the authors.

3. The infinite dilution mentioned by the authors in the manuscript is impossible to achieve in practice, and the authors propose 0.1 μL, is there any basis for this, can it meet the standard of infinite dilution?

4. The authors describe that “Table 3 clearly showed that the standard free energy of adsorption (−∆??0) decreases for every probe when the temperature increases (Figure 1) and decreasing at the same time the adsorption of molecules.” However, from Table 3, The datas of “Pentane, Hexane” are increasing. Please check the data for narrative errors, and if correlation analysis was used to fetch the data, please specify.

5. The conclusion of this manuscript is too long and not in the form of a research paper.. Please re-condense the conclusion to no more than four items, and do not show icons, formulas, etc. in the conclusion.

Author Response

Dear Editor,

Please find attached the reply to the first referee.

With my best regards

Reviewer 2 Report

In the manuscript entitled “New physicochemical methodology for the determination of the surface thermodynamic properties of solid particles” (Manuscript ID: appliedchem-2374726), the author describes a potential route to the determination of surface thermodynamic properties of alumina particles. The manuscript concludes that the new methodology based on Hamieh model provides more accurate results on the physicochemical properties of the particle surfaces.

Here are some comments and suggestions from this reviewer.

1. The manuscript takes alumina particles as an example. Is it logically the same to the other particles? Also, the detailed characterization of the particles is missing.

2. The introduction of this manuscript is too general, some more previous studies regarding the basic ideas and mechanisms should be mentioned in this section. Also, the section 4 (Discussion) should be moved to the introduction part.

3. It is suggested to use some latest papers to support this study. The only latest reference is a paper by the same author of this manuscript.

4. This manuscript involves too many classical thermodynamic equations. Is it possible to only keep the key equations in this work?

Therefore, this reviewer recommends reconsideration of the manuscript after major revision.

Moderate editing of English language

Author Response

Dear Editor,

Please find attached the reply to referee 2.

With my best regards

Reviewer 3 Report

The current work has merit but needs some polishing before publication can be guaranteed

Infinite dilution: perhaps good to mention concentration range.

Section 2.2.1 needs to be extended

Is Section 2.2.3 now new work? General comment: please highlight the novelty in mathematics more: perhaps also add in the introduction a typical (conceptual) result of the state of the art vs what the reader can expect.

Table 3: any references needed?

Figure 2 needs more statistics in view of the global average

General comment: several tables or figures could go to SI

The conclusions are in mixed abstract, “small paper”, real conclusion style

Author Response

Dear Editor,

Please find attached the reply to referee 3.

With my best regards

Round 2

Reviewer 2 Report

The revised manuscript can be accepted in the present form. 

 Minor editing of English language suggested.